# SITE: Bridging Text and Image Modalities with LLMs for 3D Scene Understanding

## Abstract

It is a fundamental challenge for embodied agents to understand and interact with complex 3D scenes. Large language models (LLMs) have demonstrated strong capabilities in text and 2D image understanding. However, existing LLMs with 3D encoders suffer from insufficient paired 3D data for scalable training. In this work, we propose **S**ingle-**I**mage and **T**ext **E**ncoders (**SITE**), a general framework using a 1D text encoder and a 2D image encoder for structured scene parsing and 3D scene understanding. Specifically, we *i*) design Scene2Text module to extract instance-level relations, *ii*) transform multi-view observations into BEV images for interpreting spatial relations, and *iii*) fuse such 1D and 2D encoders into LLM fine-tuning for consistent 3D understanding. In addition, we introduce InPlan3D, a long-sequence planning benchmark to further evaluate the embodied reasoning ability. Extensive experiments demonstrate the effectiveness and efficiency of SITE on multiple 3D scene understanding datasets and InPlan3D with less token cost. Code and dataset will be publicly released.

## 1 Introduction

3D scene understanding is a crucial task of embodied AI with broad applications in robotics, augmented reality, and autonomous systems (Chen et al., 2024a). It requires agents to perform complex operations in real-world environment (Shridhar et al., 2020). Previous methods achieve promising accuracy in single 3D tasks, *e.g.* visual grounding and semantic segmentation tasks. However, they lack the ability of general-understanding. Recent studies (Hong et al., 2023; Chen et al., 2024b; Qi et al., 2025; Huang et al., 2023d; Zheng et al., 2024; Huang et al., 2023a; Zhu et al., 2024b) focus on fine-tuning Large Language Models (LLMs) to advance 3D scene understanding, developing general-purpose assistants. These approaches incorporate the features of detected objects, constructing scene-level 3D representations by integrating multiple techniques: harnessing point cloud feature or lifting multi-view image features into 3D space.

The integration of LLMs with 3D scenes understanding enables LLMs to describe and reason in real-word environments. However, bridging 3D scenes and language presents unique challenges: 1) LLMs are predominantly trained on paired image-text data from the internet, yet the 2D visual knowledge falls short of capturing the complexity inherent in 3D scenes (Zheng et al., 2024), *ii*) richly annotated 3D data (*e.g.*, depth maps and point clouds) suitable for fine-tuning LLMs remain severely scarce (Zheng et al., 2025), *iii*) the representation of point clouds or video inputs will bring huge token overhead, which consumes lots of resources and slowing down inference. These limitations constrain the performance potential of LLMs on tasks, *e.g.* 3D scene understanding and embodied task planning (Jia et al., 2024).Given the aforementioned challenges, a fundamental question arises: *Can we propose an efficient and general solution for 3D scene understanding?*

In this paper, we introduce **S**ingle-**I**mage and **T**ext **E**ncoders (**SITE**), a general framework for structured scene parsing and 3D scene understanding, bridging text and image modalities with LLMs. It transforms 3D scenes into structured textual descriptions while efficiently capturing context information using Scene2Text module, which parses 3D scenes into two core components automatically. First, identifying the categories and geometric properties of objects. Then modeling spatial relationships by forming a 3D scene graph (§3.1 & §3.2). The well-structured 3D information will be fed into LLMs by downstream fine-tuning, enabling it to acquire the capability to parse information and perform downstream tasks (§3.3).

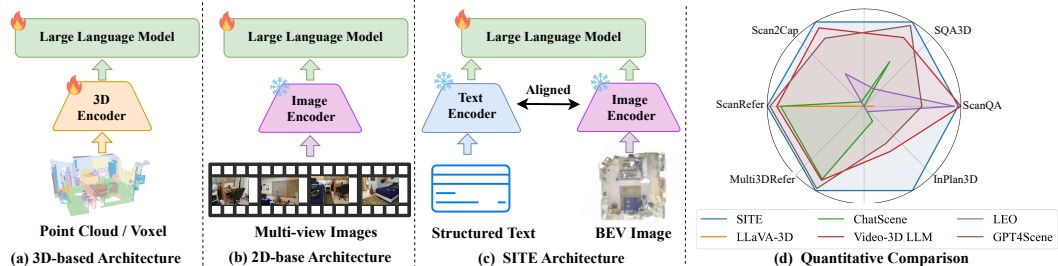

Figure 1: Overview of SITE. Previous LLM-based work achieve 3D scene understanding by either (a) training dedicated 3D encoders to map point clouds or voxels into textual space (Zhu et al., 2024a; Huang et al., 2023b;d), or (b) leveraging pretrained image encoders to capture spatial information through multi-view images or video streams (Zheng et al., 2024; Qi et al., 2025). (c) SITE exploits structured text and BEV images to process scene information, and (d) we provide performance comparison between SITE and other methods.

SITE offers several advantages: *i*) it is compact, reducing memory requirements compared with traditional methods, *ii*) the textual description is complete and interpretable. SITE achieves strong performance across different downstream tasks with less training costs. In addition, to comprehensively evaluate embodied task-planning capabilities in 3D scenes, we curate a new benchmark, InPlan3D, consisting of 3, 174 long-term planning tasks across 636 indoor scenes. Our approach achieves state-of-the-art performance on multiple 3D scene understanding datasets, *e.g.* SQA3D (Ma et al., 2022), Multi3DRefer (Zhang et al., 2023) , ScanRefer (Chen et al., 2020) and InPlan3D, compared with recent 3D LLMs (§4.1). We provide ablation studies on SITE to verify effectiveness (§4.2).

## 2 RELATED WORK

**3D Scene Understanding.** In the rapidly progressing domain of 3D scene understanding, language has emerged as a powerful tool for conveying contextual cues and formulating user intent. Core tasks including (1) 3D Visual Grounding (Chen et al., 2020; Zhang et al., 2023; Chen et al., 2023; Wang et al., 2023b; Zhao et al., 2021; Wang et al., 2023c; Unal et al., 2024), which aims to localize target objects in 3D space based on textual descriptions; (2) 3D Question Answering (3D QA) (Azuma et al., 2022; Parelli et al., 2023; Ma et al., 2022), which addresses scene-level reasoning and information retrieval through natural language queries; (3) 3D Dense Captioning (Chen et al., 2021; Yuan et al., 2022; Jiao et al., 2022; Chen et al., 2023; 2024c; Cai et al., 2022; Chen et al., 2022a), which requires generating fine-grained object-level captions with accurate spatial localization. Traditional models (Zhu et al., 2023; Jin et al., 2023) often depend on dedicated task-specific heads, which constrain their flexibility in adapting to open-ended user-assistant interactions and limits their broader applicability in general-purpose multi-modal reasoning.

**3D LLMs.** Recent efforts have increasingly focused on integrating 3D scene information into large language models (LLMs) to advance 3D scene understanding (Chen et al., 2023; 2024c;b; Fu et al., 2024; Guo et al., 2023; Hong et al., 2023; Wu et al., 2023; Fan et al., 2024; 2025). 3D-LLM (Hong et al., 2023) initially leverages rendered 2D views as input to LLMs. Methods like Chat3D (Huang et al., 2023a), LEO (Huang et al., 2023d), and ChatScene (Huang et al., 2023b) rely on off-the-shelf 3D detectors to generate object proposals, which are then integrated into language models. Meanwhile, GPT4Scene (Qi et al., 2025) captures object-level and scene-level semantic features by leveraging multi-view images and rendered BEV images, respectively. Similarly, Video 3D LLM (Zheng et al., 2024) introduces a novel paradigm that implicitly embeds 3D spatial information into video representations, eliminating the need for specialized 3D encoders. However, directly feeding scene point clouds or using multi-view images introduces longer token sequences, resulting in high training costs. Moreover, inconsistencies in cross-modal representations limit the spatial reasoning capability. To address these, SITE can capture object attributes and 3D spatial relations.

**Multimodal Embodied Tasks.** The emergence of general-purpose models exhibit strong performance across a wide range of multi-modal tasks (Lu et al., 2022; 2024; Wang et al., 2023a; Kirillov et al., 2023; Achiam et al., 2023; Hurst et al., 2024; Kim et al., 2024; Liu et al., 2023; 2024a), while they still face challenges when deployed in embodied scenarios that require perception, planning,

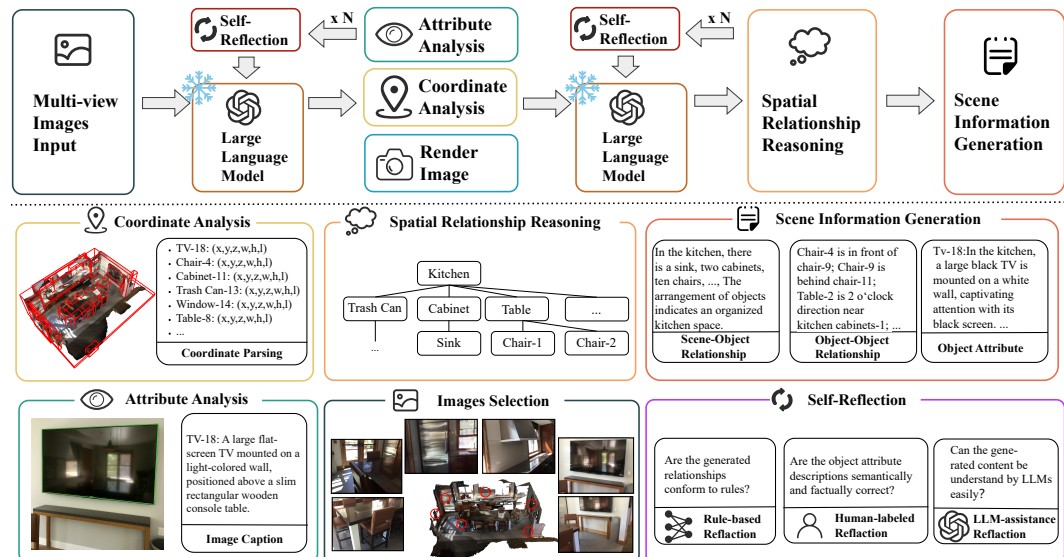

Figure 2: Illustration of Scene2Text module (§ 3.1 & § 3.2). It first identifies the categories and geometric properties of objects. Then analysis the salient spatial relations among them by forming an intermediate 3D scene graph. We insert self-reflection mechanism to provide feedback to LLMs.

and interaction with the environment (Liu et al., 2024b). To address this gap, various approaches have been proposed (Ahn et al., 2022; Huang et al., 2022b; 2023d). We propose InPlan3D, a more diverse and general-purpose benchmark dataset designed to evaluate the quality of task generation. Tasks in InPlan3D are constructed from the perspective of a indoor service robot, aiming to explore the potential of empowering embodied robot.

# 3 METHOD

In this section, we detail the SITE framework. We first introduce the scene parsing algorithm of Scene2Text module(§3.1). We further illustrate how Scene2Text generates Scene Information (§3.2). We then describe how the results parsed by Scene2Text are incorporated into the SITE framework. (§3.3). Implementation details are provided for reference (§3.4).

## 3.1 SCENE PARSING ALGORITHM

**Image Sampling.** Given a raw egocentric video, whose each frame captures a portion of the 3D scene, we first randomly select $n$ frames $\mathcal{V} = \{I_1, I_2, \ldots, I_n\}$ with corresponding camera extrinsics $\varepsilon = \{E_1, E_2, \ldots, E_n\}$. Then we could reconstruct 3D point clouds $\mathcal{P}$: $\mathcal{P} = \mathcal{R}(\{(I_i, E_i)\}_{i=1}^{N})$, where $\mathcal{R}$ represents images to point cloud projection using 3D reconstruction.

**Spatial Relationship Reasoning.** To successfully recognize diverse objects within a scene, reason about their spatial relationships, and perform task planning and execution, an agent must possess strong scene semantic understanding capabilities. We could obtain instance masks $\mathcal{M} = \{M_1, M_2, \ldots, M_K\}$ by applying 3D instance segmentation methods (*e.g.*, Mask3D (Schult et al., 2023)), where $K$ denotes the total number of objects in the scene. As shown in Algorithm 1, we propose a spatial relationship reasoning framework that integrates geometric proximity, camera-view-based inference and semantic priors to generate fine-grained and interpretable object-level spatial relations. As shown in Figure 3, it analyzes spatial

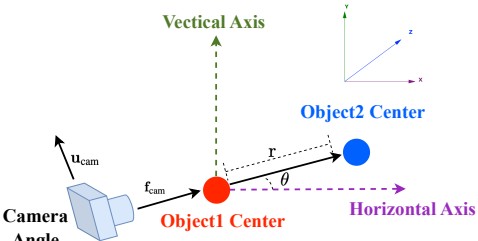

Figure 3: An example of calculating the angle between two objects. $f_{cam}$ denotes the forward direction of the camera.

**Algorithm 1** Spatial Relationship Reasoning

**Require:** Objects $A$, $B$ with centroids $\mathbf{p}_A, \mathbf{p}_B \in \mathbb{R}^3$, sizes $\mathbf{s}_A, \mathbf{s}_B \in \mathbb{R}^3$; camera forward $\mathbf{f}_{\text{cam}}$; horizon vector $\mathbf{x}$; proximity factor $\beta$, tolerance $\theta_{\text{tol}}$; semantic prior $\mathcal{R}_{\text{prior}}$

**Ensure:** Spatial relationship set $\mathcal{R}_{rel} \in \emptyset$

1: **if** $(A, B) \in \mathcal{R}_{\text{prior}}$ **then**
2:    **return** relation from $\mathcal{R}_{\text{prior}}$
3: **else if** $\|\mathbf{p}_A - \mathbf{p}_B\| < \beta \times \max(\|\mathbf{s}_A\|, \|\mathbf{s}_B\|)$ **then**
4:    Update distance $\mathcal{R}_{rel} \leftarrow \mathcal{R}_{rel} \cup \{\text{nearby}\}$
5: $\mathbf{r} \leftarrow \mathbf{p}_A - \mathbf{p}_B$ , $v \leftarrow \mathbf{r} \cdot \mathbf{f}_{\text{cam}}, \theta \leftarrow \arccos\left(\frac{\mathbf{r} \cdot \mathbf{x}}{\|\mathbf{r}\|\|\mathbf{x}\|}\right)$
6: **if** $v > 0$ **then**
7:    Update vertical $\mathcal{R}_{rel} \leftarrow \mathcal{R}_{rel} \cup \{\text{is above}\}$
8: **else**
9:    Update vertical $\mathcal{R}_{rel} \leftarrow \mathcal{R}_{rel} \cup \{\text{is below}\}$
10: **if** $\theta < \theta_{\text{tol}}$ **then**
11:    Update horizontal $\mathcal{R}_{rel} \leftarrow \mathcal{R}_{rel} \cup \{\text{in front of}\}$
12: **else**
13:    Update horizontal $\mathcal{R}_{rel} \leftarrow \mathcal{R}_{rel} \cup \{\text{left of / right of}\}$
14: Partition $[0°, 360°)$ into $N$ sectors, $k \leftarrow \text{sector}(\theta, N)$
15: Update angular $\mathcal{R}_{rel} \leftarrow \mathcal{R}_{rel} \cup \{k \text{ o'clock}\}$

**Algorithm 2** Self-Reflection Mechanism

**Require:** Captions $\mathcal{C}$ and relationship $\mathcal{R}$; marked images $\mathcal{I}$; advanced LLM $\mathcal{M}$; value function $\mathcal{V}$; threshold $\tau$; GT label $\mathcal{G}$

**Ensure:** Refined captions $\hat{\mathcal{C}}$, relationship $\hat{\mathcal{R}}$

1: $\hat{\mathcal{C}} \leftarrow \mathcal{C}$;    $\hat{\mathcal{R}} \leftarrow \mathcal{R}$
2: **for all** object caption $c_i \in \hat{\mathcal{C}}$ **do**
3:    Formulate QA prompt $\mathcal{Q}_i$ from $c_i$
4:    Obtain predicted index $o_i \leftarrow \mathcal{M}(\mathcal{Q}_i)$
5:    $s_i \leftarrow \text{Accuracy}(o_i, \mathcal{G})$
6:    **if** $s_i < \tau$ **then**
7:      Obtain correction $c_i' \leftarrow \mathcal{V}(c_i)$
8:      Update $\hat{\mathcal{C}} \leftarrow (\hat{\mathcal{C}} \setminus \{c_i\}) \cup \{c_i'\}$
9: **for all** relation $r_j \in \hat{\mathcal{R}}$ **do**
10:    Compose input pair $(\mathcal{I}, r_j)$
11:    $s_j \leftarrow \mathcal{M}((\mathcal{I}, r_j))$
12:    **if** $s_j < \tau$ **then**
13:      $r_j' \leftarrow \mathcal{V}(r_j)$
14:      Update $\hat{\mathcal{R}} \leftarrow (\hat{\mathcal{R}} \setminus \{r_j\}) \cup \{r_j'\}$
15: **return** $\hat{\mathcal{C}}$, $\hat{\mathcal{R}}$

relationships between objects by jointly considering the camera angle, viewing distance and other relevant factors. For the detailed procedure for relationship computation, please refer to Appendix B.

**Instance Projection.** When processing a query like *find a table directly in front of the black armchair*, the model must first identify key attributes such as *black*, *armchair*, and *table*, inferring based on the combined attribute and spatial cues. This capability is fundamental for supporting downstream tasks *e.g.* object localization, detailed description generation, and high-level task planning.

We project the 3D bounding box of a given object onto multi-view images. The corresponding image regions are then cropped based on the projected bounding boxes and processed using BLIP-2 (Li et al., 2023) to generate multiple brief captions for the corresponding object. Next, we apply CLIP (Radford et al., 2021) to compute the similarity between each cropped image and its brief caption, thereby assessing textvisual alignment. Among all generated captions, we select the top 10 sentences with the highest CLIP similarity scores. The candidate sentences are subsequently passed to an advanced large language model (*e.g.*, GPT-4o (Hurst et al., 2024)) to integrate and refine the outputs from BLIP-2.

## 3.2 SCENE-TO-LANGUAGE TRANSLATION

**Scene Information Generation.** After parsing the attribute and positional relationships of each object, we employ an advanced LLM (*e.g.*, GPT-4o (Hurst et al., 2024)) to generate structured descriptions, which comprises three components: 1) System Message that instructs the LLM about the structure of the inputs; 2) Object Caption section, wherein the attributes of each object are described in language; 3) Relationship Generation component serializes the scene graph, represented as $\{\text{obj}_1, \text{obj}_2, \text{rel}\}$ triplets, into coherent natural language expressions suitable for LLM processing.

**Self-Reflection.** As shown in Figure 2, after initially generating object-level captions and inter-object relationships, it is crucial to perform reflective analysis and targeted optimization to ensure the textual content faithfully represents the 3D scene. We observe that direct translations from visual features or coordinate data may occasionally result in incomplete or ambiguous descriptions, especially in spatially dense environments. To address this, we introduce a refinement pipeline that incorporates spatial priors, context-aware consistency checks, and human-in-the-loop feedback where necessary. As detailed in Algorithm 2, given the object ID to be evaluated, we first retrieve the corresponding Caption $\mathcal{C}$ and Spatial Relationship $\mathcal{R}$ from the Scene Information. These textual descriptions are then fed into a pretrained **Value Function** $\mathcal{V}$, which jointly considers the marked multi-view images to assign a quality score. More details are included in Appendix C.

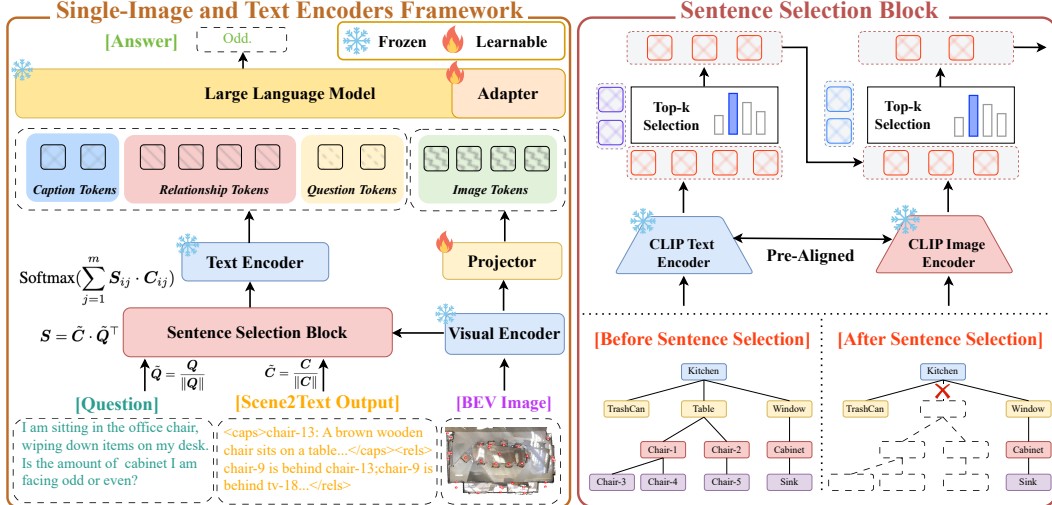

Figure 4: The architecture of SITE (§3.3). Given object-level textual captions, the Sentence Selection Block computes token-level similarity scores between the captions and the questions. Then selects the most semantically relevant tokens with respect to the question intent. The combined representation is subsequently fed into the LLM to generate context-aware answers.

## 3.3 3D SCENE UNDERSTANDING WITH SENTENCE SELECTION

**Multimodal Reasoning Framework.** Our goal is to extend pre-trained LLMs for textual scene information inputs. We leverage scene-to-language translation framework for multimodal alignment (§3.2), which parses semantic information and spatial relationship between objects, eliminates the need for heavy multi-modal alignment. As shown in Figure 4, to enable LLMs to effectively utilize textual scene information represented, we design the following tasks to facilitate alignment between scene representations and model understanding: *i) Scene-level caption*: Given the text-driven representation, generate a brief caption about the indoor scene, *ii) Spatial relationship reasoning*: Given the text-driven representation and question, predict the answer.

**Sentence Selection Block.** For LLMs, the textual information in §3.2 is overly verbose, which may hinder efficient context comprehension and reasoning. To address this issue, we introduce a sentence selection mechanism that filters out irrelevant content and preserves question-relevant information, thereby enhancing the model's capacity for grounded scene understanding. During the text-only pre-alignment stage, the input consists of a set of object captions from scene information with related questions. These textual components are first encoded using a frozen CLIP Text Encoder within the Sentence Selection Block, which represents each sentence as a sequence of token embedding. Then computes the cosine similarity between the question embedding and each caption embedding to assess their semantic relevance. Based on the computed similarity scores, the top-$k$ most relevant caption tokens are selected. Detailed computation process is shown in Equation 1:

$$\tilde{Q} = Q/\|Q\|, \ \tilde{C} = C/\|C\|, \ S = \tilde{C} \cdot \tilde{Q}^\top, \ \alpha_i = \text{Softmax}\big(\sum_{j=1}^{m} S_{ij} \cdot C_{ij}\big), \qquad (1)$$

where $Q \in \mathbb{R}^{n \times d}$ is the question token embeddings, $C \in \mathbb{R}^{m \times d}$ is the caption token embeddings, $n$ and $m$ is the number of question and caption tokens, respectively; $\tilde{Q}, \tilde{C}$ is L2-normalized token embeddings, $S \in \mathbb{R}^{m \times n}$ is pair-wise cosine similarity matrix between caption and question tokens, $\alpha \in \mathbb{R}^m$ is normalized attention weights over caption tokens. The top-$k$ caption tokens with the highest $\alpha_i$ scores are selected as the most semantically relevant context for the question. During the fine-tuning phase, we repeat the above process by replacing the matrix $Q$ with image feature embeddings extracted by the CLIP Image Encoder, and replacing $C$ with the caption token embeddings obtained from the first-round selection. The image features are then used to further refine and filter the caption tokens to better align with the scene context. After passing through the Sentence Selection Block, we can select the top-$k$ objects and their corresponding captions from the original pool of over 60 objects and captions, reducing token consumption. Also, we can also improve the performance by providing BEV images (Qi et al., 2025). Following the pretraining-finetuning paradigm, the pre-aligned LLMs can be fine-tuned with BEV images' input for improved downstream task per-

Figure 5: Illustration of proposed InPlan3D benchmark (§4.1). InPlan3D benchmark emphasizes the model's ability to solve problems step-by-step, requiring problem define, object retrieval, and action planning abilities. Differ from previous tasks, InPlan3D focuses on evaluating reasoning abilities, pushing forward the development of general-purpose, human-aligned embodied agents.

formance. Comparative experiment in Appendix A shows that BEV-assisted approach can improve performance in QA and Caption tasks, while the improvement in grounding tasks is limited.

**Loss function.** To standardize the training process, all tasks are reformulated into a unified user-assistant interaction format. Consequently, during the joint training phase, the model is optimized solely using the Cross-Entropy loss from the language modeling objective. The goal is to learn the trainable parameters $\theta$ by minimizing the negative log-likelihood of the assistants target response textual sequence $t^{\text{res}}$. Given the input prefix textual sequence $t^{\text{prefix}}$ which includes system messages, the top-k selected scene information and user instructions, the loss function is defined in Equation 2:

$$\mathcal{L}(\theta) = -\sum_{i=1}^{k} \log P\big(t_i^{\text{res}} \mid t_{[1,\ldots,i-1]}^{\text{res}}, t^{\text{prefix}}\big), \tag{2}$$

where $k$ is the number of tokens in the response sequence, and $t_{[1,\ldots,i-1]}^{\text{res}}$ denotes the sequence of the previous $i-1$ tokens in the response. The set of trainable parameters $\theta$ represents the visual projector, a 3-layer-MLP, as long as LLM adapter.

## 3.4 IMPLEMENTATION DETAILS

We leverage Qwen2-7B as the LLM backbone. For better performance, we utilize the text tokenizer and the ViT (Wang et al., 2023a) from Qwen2-VL (Wang et al., 2024) as the multi-modal encoder. In Sentence Selection Block, we use pretrained CLIP-ViT-L (Radford et al., 2021) as the multi-modal encoder. We use LoRA (Hu et al., 2022) for supervised fine-tuning with a rank of $8$. During training, the text tokenizer, ViT, CLIP, and LLM backbone are frozen, and the projector and additional adapter for LLM is trainable. We train the model on a mixture of tasks comprising scene-level caption and spatial relationship reasoning. SITE is trained on four 80G-A800 GPUs in 13 hours. More details of SITE and baselines are included in Appendix D. The implementation will be released.

## 4 EXPERIMENT

**Datasets.** We conduct experiments on six different benchmarks across $1,513$ scenes: ScanQA (Azuma et al., 2022) and SQA3D (Ma et al., 2022) for visual question answering, Scan2Cap (Chen et al., 2021) for dense captioning, ScanRefer (Chen et al., 2020) for single-object visual grounding, and Multi3DRefer (Zhang et al., 2023) for multi-object visual grouding. In addition, we propose InPlan3D, a benchmark to evaluate the model's capability in indoor task planning based on ScanNet (Dai et al., 2017). In Figure 5, existing benchmarks (*e.g.*, 3D Quesion Answering, 3D Visual Grounding) mainly focus on specific tasks, with single-turn dialogue format. In contrast, InPlan3D incorporates multi-task and multi-turn reasoning dialogue, requiring the model to understand and reason over complex environments (see more details in Appendix E).

**Metrics.** Following existing methods (Huang et al., 2023d;a; Qi et al., 2025), we assess accuracy using Acc@0.25 and Acc@0.5 for ScanRefer (Chen et al., 2020) with IoU thresholds of 0.25 and 0.5. For Multi3DRefer (Zhang et al., 2023), we employ a F1 score at IoU thresholds of 0.25 and 0.5. For Scan2Cap (Chen et al., 2021), we utilize CIDEr@0.5 and BLEU-4@0.5. For ScanQA (Azuma et al.,

Table 1: **Comparison with baselines.** *Task-specific Models* are customized for specific tasks through task heads. Point and Vision Encoders correspond to input modalities.

| Method | Point Encoder | Vision Encoder | ScanRefer | | Multi3DRef | | Scan2Cap | | ScanQA | | SQA3D |
|---|---|---|---|---|---|---|---|---|---|---|---|
| | | | Acc@0.25 | Acc@0.5 | F1@0.25 | F1@0.5 | B-4@0.5 | C@0.5 | C | EM | EM |
| *Task-specific Models* | | | | | | | | | | | |
| ScanRefer (Chen et al., 2020) | ✓ | ✗ | 37.3 | 24.3 | – | – | – | – | – | – | – |
| MVT (Huang et al., 2022a) | ✓ | ✗ | 40.8 | 33.3 | – | – | – | – | – | – | – |
| 3DVG-Trans (Zhao et al., 2021) | ✓ | ✗ | 45.9 | 34.5 | – | – | – | – | – | – | – |
| ViL3DRel (Chen et al., 2022b) | ✓ | ✗ | 47.9 | 37.7 | – | – | – | – | – | – | – |
| M3DRef-CLIP (Zhang et al., 2023) | ✓ | ✗ | 51.9 | 44.7 | 42.8 | – | 38.4 | – | – | – | – |
| Scan2Cap (Chen et al., 2021) | ✓ | ✗ | – | – | – | – | 22.4 | 35.2 | – | – | – |
| ScanQA (Azuma et al., 2022) | ✓ | ✗ | – | – | – | – | – | – | 64.9 | 21.1 | 47.2 |
| 3D-VisTA (Zhu et al., 2023) | ✓ | ✗ | 50.6 | 45.8 | – | – | 34.0 | 66.9 | 69.6 | 22.4 | 48.5 |
| *3D LLMs* | | | | | | | | | | | |
| 3D-LLM(Flamingo) (Hong et al., 2023) | ✓ | ✓ | 21.2 | – | – | – | – | – | 59.2 | 20.4 | – |
| 3D-LLM(BLIP2-flant5) (Hong et al., 2023) | ✓ | ✓ | 30.3 | – | – | – | – | – | 69.4 | 20.5 | – |
| Chat-3D (Wang et al., 2023d) | ✓ | ✗ | – | – | – | – | – | – | 53.2 | – | – |
| Chat-3D v2 (Huang et al., 2023c) | ✓ | ✗ | 42.5 | 38.4 | 45.1 | 41.6 | 31.8 | 63.9 | 87.6 | – | 54.7 |
| LL3DA (Chen et al., 2024b) | ✓ | ✗ | – | – | – | – | 36.0 | 62.9 | 76.8 | – | – |
| SceneLLM (Fu et al., 2024) | ✓ | ✗ | – | – | – | – | – | – | 80.0 | 27.2 | 53.6 |
| LEO (Huang et al., 2023d) | ✓ | ✓ | – | – | – | – | 38.2 | 72.4 | 101.4 | 24.5 | 50.0 |
| Grounded 3D-LLM (Chen et al., 2024d) | ✓ | ✗ | 47.9 | 44.1 | 45.2 | 40.6 | 35.5 | 70.6 | 72.7 | – | – |
| PQ3D (Zhu et al., 2024b) | ✓ | ✓ | 57.0 | 51.2 | – | 50.1 | 36.0 | 80.3 | – | – | 47.1 |
| ChatScene (Huang et al., 2023b) | ✓ | ✓ | 55.5 | 50.2 | 57.1 | 52.4 | 36.3 | 77.1 | 87.7 | 21.6 | 54.6 |
| LLaVA-3D (Zhu et al., 2024a) | ✓ | ✓ | 54.1 | 42.4 | – | – | 41.1 | 79.2 | 91.7 | 27.0 | 55.6 |
| GPT4Scene (Qi et al., 2025) | ✗ | ✓ | 62.6 | 57.0 | 64.5 | 59.8 | 40.6 | 79.1 | 96.3 | 26.5 | 60.6 |
| Video-3D LLM (Zheng et al., 2024) | ✗ | ✓ | 58.1 | 51.7 | 58.0 | 52.7 | 41.3 | 83.8 | **102.1** | **30.1** | 58.6 |
| SITE (text-only) | ✗ | ✗ | 59.3 | 53.6 | 63.1 | 58.7 | 36.8 | 80.0 | 89.5 | 22.9 | 57.7 |
| SITE | ✗ | ✓ | **64.5** | **59.4** | **66.1** | **60.7** | **41.7** | **84.1** | 93.7 | 23.4 | **61.2** |

2022), CIDEr (Vedantam et al., 2015) and BLEU-4 (Papineni et al., 2002) are used. SQA3D (Ma et al., 2022) is evaluated using exact match accuracy (EM) and its refined version, EM-R.

## 4.1 COMPARISON WITH STATE-OF-THE-ART METHODS

**Baselines.** Based on the architecture, baselines can be categorized into task-specific models and 3D LLMs. Traditional task-specific models are typically designed for individual tasks and require separate training on corresponding datasets. In contrast, 3D LLMs are generally capable of handling multiple indoor scene understanding tasks simultaneously. 3D LLMs are trained on various datasets covering diverse tasks, eliminating task-specific design or fine-tuning for each individual task.

- **Task-Specific Models:** Models such as ScanRefer (Chen et al., 2020) and ScanQA (Azuma et al., 2022) establish initial benchmarks for the ScanRefer and ScanQA datasets, respectively. 3D-VisTA (Zhu et al., 2023) aim to develop versatile 3D visual-language frameworks by focusing on pre-training strategies for 3D scene-language alignment. M3DRef-CLIP (Zhang et al., 2023) introduces multi-object grounding, enhancing single-object grounding performance. ConcreteNet (Unal et al., 2024), the state-of-the-art model on ScanRefer (Chen et al., 2020), innovates three methods to augment verbal-visual fusion for 3D visual grounding.

- **3D LLMs:** 3D-LLM (Hong et al., 2023) utilizes location tokens for object grounding but is constrained by data scarcity. LL3DA (Chen et al., 2024b) and SceneLLM (Fu et al., 2024) processes point clouds directly, responding to textual instructions and visual prompts. Grounded 3D-LLM (Chen et al., 2024d), PQ3D (Zhu et al., 2024b), and LLaVA-3D (Zhu et al., 2024a) achieve strong performance on 3D visual grounding tasks by joint training with a 3D detection module. Chat3D (Huang et al., 2023a), LEO (Huang et al., 2023d) and ChatScene (Huang et al., 2023b) integrate visual and point cloud modalities, using language as guidance to facilitate cross-modal fusion and understanding. GPT4Scene (Qi et al., 2025) apply multi-view images and marked BEV images as input, while Video-3D LLM (Zheng et al., 2024) uses videos.

**Performance on Scene Understanding.** In Table 1, SITE outperforms all existing task-specific models and 3D LLM baselines on most tasks, demonstrating the strength of our text-driven framework for 3D scene understanding. The term *text-only* indicates that LLM has only textual description as input. Compared with current state-of-the-art 3D LLMs that rely heavily on visual inputs, our method requires significantly fewer image inputs with superior performance across downstream

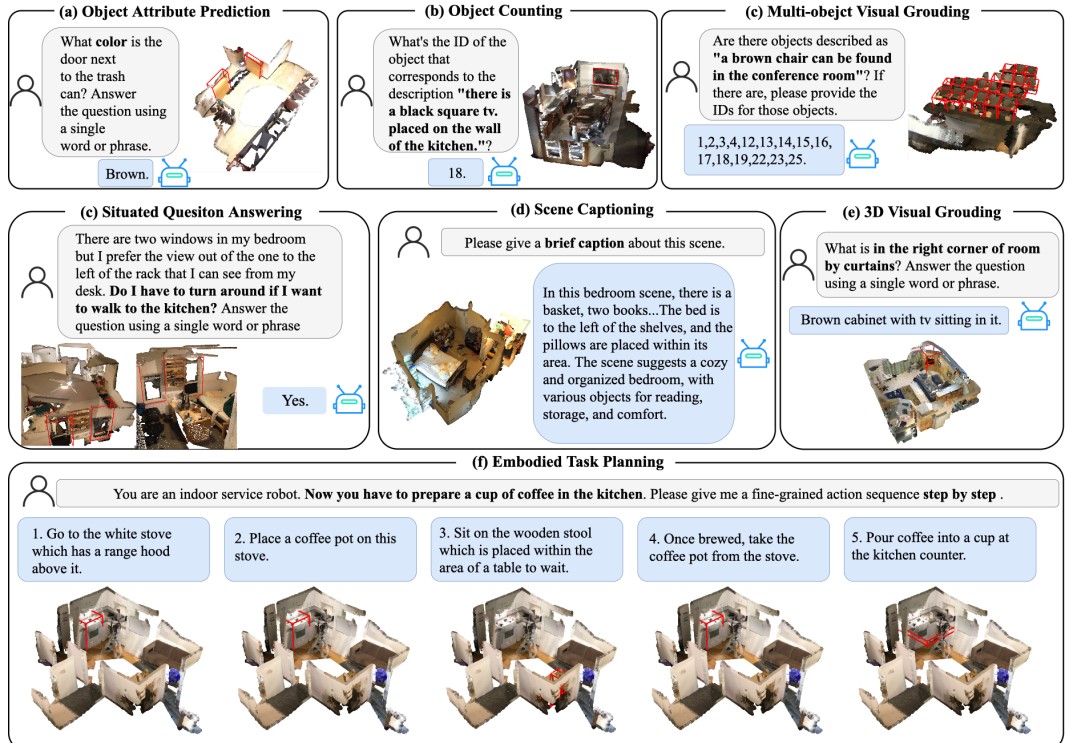

Figure 6: Visualization of various downstream tasks, including 3D dense captioning, 3D question answering (QA), 3D visual grounding, and embodied task planning. White boxes indicate user instructions, while green boxes present the responses generated by SITE.

tasks. For visual grounding, SITE achieves a new state-of-the-art with 64.5% Acc@0.25 and 59.4% Acc@0.5 on ScanRefer, and 60.7% F1@0.5 on Multi3dRefer, verifying the efficacy of our text-driven scene parsing and object selection strategies. For 3D VQA tasks, our model attains top performance on both 91.3% CIDEr and 24.6% EM on ScanQA and 61.2% EM on SQA3D, confirming the model's ability to understand, ground, and reason over complex 3D scenes.

**Performance on InPlan3D.** In Table 2, we report the performance of the 3D LLMs using the same splits and initial conditions. The term *text-only* indicates that LLM has only textual description as input. More calculation details of $G_{\text{Acc}}$ and $T_{\text{Acc}}$ are shown in Appendix E. SITE achieves state-of-the-art performance across multiple evaluation dimensions. Despite using only a single BEV image as visual input, significantly fewer than prior 3D LLMs that depend on large-scale multi-view inputs, SITE surpasses most baselines in both task-level and step-level accuracy. Specifically, SITE achieves the highest $G_{\text{Acc}}$ (47.23%) and $T_{\text{Acc}}$ (65.91%), demonstrating strong task grounding and execution ability. Furthermore, in terms of language similarity, our method leads all competitors with top scores across all metrics. Notably, SITE outperforms vision-heavy methods like GPT4Scene (Qi et al., 2025) and ChatScene (Huang et al., 2023b).

Table 2: **Comparison on Planning Task.** $G_{\text{Acc}}$ evaluates task-level accuracy, while $T_{\text{Acc}}$ reflects step-level accuracy, and language quality is measured by METEOR (Banerjee & Lavie, 2005), ROUGE (Lin, 2004), BLEU-4 (Papineni et al., 2002), and CIDEr (Vedantam et al., 2015).

| Method | Point Encoder | Vision Encoder | Task-Level $G_{\text{Acc}}$ | Step-Level $T_{\text{Acc}}$ | METEOR | ROUGE | BLEU-4 | CIDEr |
|---|---|---|---|---|---|---|---|---|
| PQ3D (Huang et al., 2023d) | ✓ | ✓ | 36.47 | 46.83 | 12.87 | 38.75 | 15.03 | 70.23 |
| LEO (Huang et al., 2023d) | ✓ | ✓ | 37.13 | 47.59 | 13.06 | 39.42 | 15.37 | 71.91 |
| ChatScene (Huang et al., 2023b) | ✓ | ✓ | 38.32 | 48.87 | 13.33 | 40.14 | 15.78 | 72.36 |
| GPT4Scene (Qi et al., 2025) | ✗ | ✓ | 41.52 | 52.45 | 13.98 | 42.28 | 16.87 | 76.71 |
| Video-3D LLM (Zheng et al., 2024) | ✗ | ✓ | 42.25 | 54.98 | 14.41 | 43.50 | 17.63 | 77.24 |
| SITE (text-only) | ✗ | ✗ | 45.69 | 58.28 | 13.79 | 41.66 | 19.12 | 74.31 |
| SITE | ✗ | ✓ | **47.23** | **65.91** | **15.04** | **44.96** | **19.87** | **80.17** |

## 4.2 ABLATION STUDY

**Effectiveness of Scene2Text module.** In Table 3, we explore the impact of different relationship generation strategies of Scene2Text on performance. The Coordinate setting directly encodes each objects 3D center coordinates and bounding box dimensions into textual descriptions, serving as a low-level representation. The Simple Relationship strategy expresses coarse spatial relations such as *in front of*, *left of*, or *above*, based solely on the camera view, without precise angular reasoning. In contrast, the Complex Relationship strategy grains angular calculations and describes spatial relations with higher precision, accounting for various directional expressions such as *to the right*, *below* or *at 4 o'clock* thereby capturing multiple plausible spatial configurations. The Complex Relationship setting achieves consistent performance improvements across different benchmarks.

Table 3: **Ablation study on Scene2Text module.** We compare different strategies for generating relational information. "Coordinate" is based on coordinates, "Simple" uses semantic relations, and "Complex" includes hierarchical and contextual relations.

| Expression Type | ScanRefer | Multi3DRef | SQA3D |
|---|---|---|---|
| | Acc@0.5 | F1@0.5 | EM |
| Coordinate | 18.4 | 14.4 | 16.5 |
| Simple | 38.9 | 34.3 | 39.6 |
| Complex | **59.4** | **60.7** | **61.2** |

**Effectiveness of Sentence Selection Block.** We validate the effectiveness of Sentence Selection block in Table 4. When no selection is applied and all captions and relationships are fed into the model (All Captions & Relationships), the performance on downstream tasks is clearly limited. Sentence Selection block achieves notable improvements across all benchmarks, while also significantly reducing average inference time to 108ms, significantly lower than GPT4Scene (562ms) and Video-3D LLM (1204ms). This setup leads to the best performance on all tasks, including ScanRefer Acc@0.5 (59.4), Multi3DRef F1@0.5 (60.7), Scan2Cap(84.1), CIDEr@0.5 (93.7), and SQA3D EM (61.2). These results confirm that combining multi-modal information through a cascaded selection strategy can significantly boost scene understanding and reasoning capabilities, while maintaining a favorable balance between performance and efficiency.

Table 4: **Ablation study on Sentence Selection.** *All Captions & Relationships* denotes that filtering is not applied to the Scene Information. *Sentence Selection* filters Scene Information based on textual input and BEV images. We also list the performance of GPT4Scene (Qi et al., 2025) and Video-3D LLM (Zheng et al., 2024) under default settings for reference.

| Method | Average Inference Time | Training Time | ScanRefer | Multi3DRef | Scan2Cap | ScanQA | SQA3D |
|---|---|---|---|---|---|---|---|
| | | | Acc@0.5 | F1@0.5 | C@0.5 | CIDEr | EM |
| All Captions & Relationships | 285 ms | 10h | 47.2 | 39.6 | 69.1 | 74.9 | 43.7 |
| Sentence Selection | **169 ms** | **6h** | **59.4** | **60.7** | **84.1** | 93.7 | **61.2** |
| GPT4Scene | 962 ms | 12h | 57.0 | 59.8 | 79.1 | 96.3 | 60.6 |
| Video-3D LLM | 1204 ms | 32h | 51.7 | 52.7 | 83.8 | **102.1** | 58.6 |

**Visualization.** Figure 6 showcases the versatile capabilities of txhe proposed SITE framework across a wide range of downstream 3D scene-language tasks. Importantly, the rendered scene images are used for visualization only and are not provided as input to the model, emphasizing the strength of SITE in understanding and reasoning over scenes using text-driven representations alone.

## 5 CONCLUSION

In this paper, we propose **SITE** that bridges 3D observation and language. By identifying key objects, attributes, and spatial relationships, Scene2Text generates rich, natural-language summaries of 3D scenes without human intervention. This text-driven representation enables holistic 3D scene understanding using only textual input, significantly reducing the reliance on dense visual data and domain-specific encoders. Experiments show that the generated descriptions are accurate, interpretable, and effective for supporting downstream tasks, such as 3D visual grounding, question answering, and dense captioning. Furthermore, to evaluate the practicality in real-world scenarios, we introduce InPlan3D, a diverse benchmark for embodied task planning in indoor environments. The results highlight the potential of leveraging language as a universal medium for 3D scene understanding, offering a scalable and efficient solution.

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

## A  ADDITIONAL ABLATION STUDY

In this section, we provide additional ablation experiments to further investigate key components of our approach. We compare the reconstruction quality of different point cloud reconstruction methods (**§A.1**). We also analyze the impact of using pre-annotated labels from the ScanNet dataset *vs.* labels generated by Mask3D on overall performance (**§A.2**).

### A.1  ABLATION STUDY ON DIFFERENT CONSTRUCTION METHODS

Deep learning-based 3D reconstruction methods (*e.g.*, VGGT Wang et al. (2025)) offer the advantage of lower computational cost, enabling direct prediction of point clouds from multi-view RGB images. In contrast, as shown in Figure 7, traditional reconstruction methods (*e.g.*, ScanNet Dai et al. (2017)) synthesize scene point clouds from RGB and depth (RGB-D) streams combined with camera intrinsics and extrinsics, resulting in higher accuracy and better reconstruction quality. In this work, we conduct scene parsing based on the point clouds provided by ScanNet. In future work, we will explore leveraging scenes reconstructed using VGGT to improve the efficiency of the Scene Information construction process.

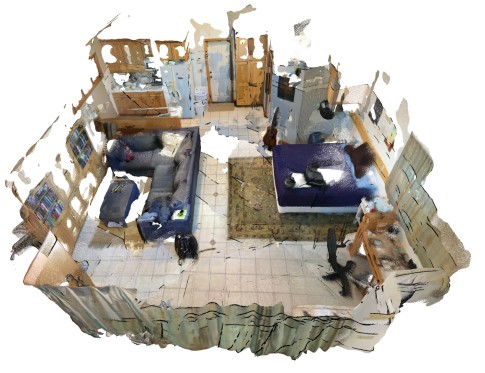
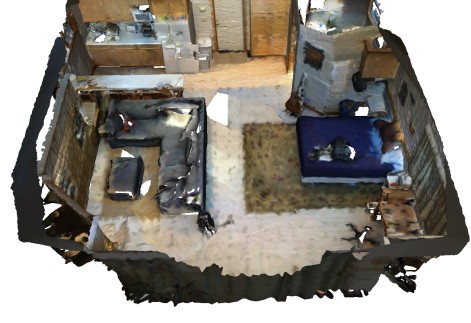

|  (a) VGGT  |  (b) RGB-D (ScanNet)  |

Figure 7: Illustrative comparison of reconstructed scenes using (a) VGGT Wang et al. (2025) and (b) RGB-D (ScanNet Dai et al. (2017)). VGGT reconstructs scenes using only 128 multi-view RGB images, offering convenience but lacking fine-grained details.

### A.2  ABLATION STUDY ON GROUND TRUTH LABELS *vs.* MASK3D LABELS

The type of labels usually has a notable impact on the construction of Scene Information, the generation strategy of model inputs, as well as evaluation metrics and implementation code. Following previous works, we adopt instance segmentation results generated by Mask3D Schult et al. (2023) as the default labeling method in this paper. In this section, we further investigate how using Ground Truth (GT) labels provided by the ScanNet dataset affects the final model performance, aiming to assess the influence of label quality on scene understanding effectiveness.

Table 5 presents an ablation study comparing the performance of the SITE framework when using different types of instance segmentation labels. We take the text-only Scene Information as input, with Qwen2-7B Wang et al. (2024) as the base model. Across all evaluated downstream tasks, the model using GT labels weakly outperforms the one using Mask3D Schult et al. (2023) predictions. Note that for the visual grounding task, using ground truth labels eliminates bounding box prediction errors. As a result, the IoU values are either 0 or 1, leading to identical scores for both @0.25 and @0.5 thresholds in the evaluation metrics. Specifically, on the ScanRefer Chen et al. (2020) task, GT labels lead to a noticeable improvement, raising Acc@0.25 from 59.3 to 63.5 and Acc@0.5 from 53.6 to 63.5. A similar pattern is observed in Multi3DRef Zhang et al. (2023), where both F1@0.25 and F1@0.5 increase from 63.1 and 58.7 to 67.2, respectively.

Table 5: Ablation Study on Ground Truth labels *vs.* Mask3D labels.

| Method | ScanRefer | | Multi3DRef | | Scan2Cap | | ScanQA | | SQA3D |
|---|---|---|---|---|---|---|---|---|---|
| | Acc@0.25 | Acc@0.5 | F1@0.25 | F1@0.5 | B-4@0.5 | C@0.5 | C | EM | EM |
| SITE (w/ Mask3D labels) | 59.3 | 53.6 | 63.1 | 58.7 | 36.8 | 80.0 | 89.5 | 22.9 | 57.7 |
| SITE (w/ GT labels) | **63.5** | | **67.2** | | **37.3** | **81.1** | **90.7** | **23.4** | **58.6** |

## B  SPATIAL RELATIONSHIP PARSING DETAILS

Firstly, we got the bounding boxes' coordinates and size from two objects: $(x_1, y_1, z_1, w_1, h_1, l_1)$ and $(x_2, y_2, z_2, w_2, h_2, l_2)$. Then we compute the Euclidean distance $d$ between two objects as following:

$$d = \sqrt{(x_1 - x_2)^2 + (y_1 - y_2)^2 + (z_1 - z_2)^2}. \tag{3}$$

Next, we propose a spatial relationship reasoning framework that integrates geometric proximity, camera view-based directional inference, and semantic priors to generate fine-grained and interpretable object-level spatial relations. To more robustly determine spatial proximity, we also take into account the sizes of the two objects. We calculate the maximum bounding box dimension, and define an adaptive proximity threshold based on a scaling factor $\beta$. If the distance $d$ is smaller than $\beta$ times the maximum of the two object sizes, the system classifies the spatial relation as ambiguous and assigns soft labels (*e.g.*, nearby). When $d$ exceeds $\beta$ times, directional reasoning is performed to determine relations such as in front of, left of, above, or an o'clock-style description. As shown in Figure 3, given the forward vector $\boldsymbol{f}_{\text{cam}}$ and the upward vector $\boldsymbol{u}_{\text{cam}}$ of the camera, we project the relative position vector $\boldsymbol{r}$ between the center of two objects onto these directional bases. We then compute the projection of $\boldsymbol{r}$ onto $\boldsymbol{u}_{\text{cam}}$ to determine whether one object is positioned above or below the other for vertical reasoning. For horizontal reasoning, we project $\boldsymbol{r}$ onto the horizontal plane orthogonal to $\boldsymbol{u}_{\text{cam}}$ and compute the deviation angle $\theta$ between $\boldsymbol{r}$ and $\boldsymbol{f}_{\text{cam}}$. If $\theta$ falls within predefined angular ranges corresponding to canonical directions (*e.g.*, in front of, behind, left of, right of), we assign discrete relational labels accordingly. However, if $\theta$ deviates beyond a specified angular tolerance $\theta_{\text{tol}}$, we adopt a finer-grained oclock-style representation. The 360-degree horizontal plane is divided into $N_{\text{o'clock}}$ equal sectors, and $\theta$ is mapped to natural language labels. In addition, we incorporate a set of semantic prior rules to account for strongly constrained object-object relationships. When a given object pair matches one of these prior templates, the semantic label from $\mathcal{R}_{\text{prior}}$ takes precedence, bypassing distance and angular reasoning.

## C  MORE DETAILS ABOUT SELF-REFLECTION MECHANISM

Figure 8 illustrates the self-reflection mechanism extensively utilized in the SITE framework to enhance the quality of Scene Information. We employ a value function to evaluate the quality of Scene Information generated by a high-level LLM. This function returns a score that determines whether a given sentence should be regenerated. Our scoring model adopts the architecture of a BLIP-2 Li et al. (2023) visual encoder and a tiny T5-3b Raffel et al. (2020) model as decoder. The T5 decoder outputs a score between 0 and 1. We train the Value Function using $1,000$ samples comprising both human annotations and LLM-generated descriptions, injecting human preferences, objective facts, and physical laws.

## D  TRAINING DETAILS

The training process of the LLM consists of two stages: textual scene information alignment and instruction fine-tuning on downstream tasks. All experiments were conducted using $4 \times 80G$ A800 GPUs with a BF16 data type. During the instruction tuning stage, we train our model for one epoch with a total batch size of 16 and a learning rate 1e-5. Throughout both stages, we employ flash-attention Dao et al. (2022), the AdamW Loshchilov & Hutter (2017) optimizer, and a cosine learning rate scheduler Loshchilov & Hutter (2016). Further details regarding hyper-parameters are documented in Table 6.

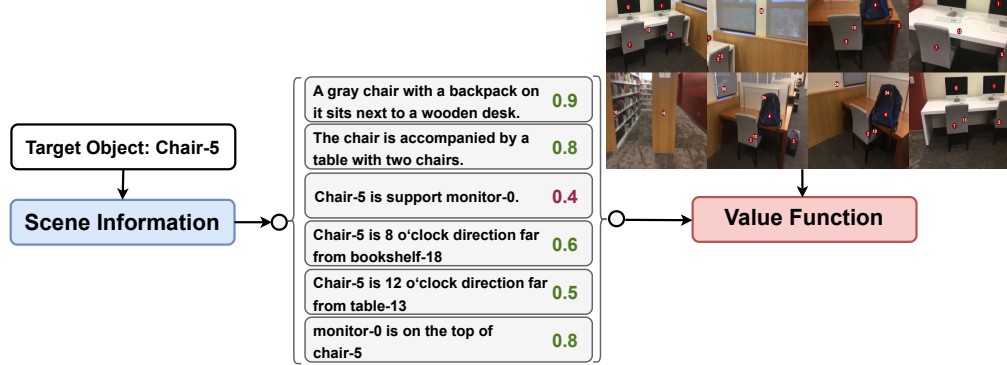

Figure 8: Illustration of the self-reflection mechanism. We mitigate hallucination issues by training a Value Function to evaluate the textual content generated by advanced LLMs, incorporating physical commonsense and human preferences into the assessment process.

Table 6: Training Hyperparameters for Three Training Stages

| Text-only Fine-tuning Stage | | Multimodal Fine-tuning Stage | |
|---|---|---|---|
| **Hyperparameter** | **Value** | **Hyperparameter** | **Value** |
| Optimizer | AdamW | Optimizer | AdamW |
| Weight decay | 0.05 | Weight decay | 0.05 |
| Betas | [0.9, 0.999] | Betas | [0.9, 0.999] |
| Learning rate | $1 \times 10^{-5}$ | Learning rate | $1 \times 10^{-5}$ |
| Warmup ratio | 0.1 | Warmup ratio | 0.1 |
| Parallel strategy | DDP | Parallel strategy | DDP |
| Type of GPUs | NVIDIA A800 | Type of GPUs | NVIDIA A800 |
| Number of GPUs | 4 | Number of GPUs | 4 |
| Batch size per GPU (total) | 4 (16) | Batch size per GPU (total) | 4 (16) |
| Training precision | bfloat16 | Training precision | bfloat16 |
| Gradient norm | 5.0 | Gradient norm | 5.0 |
| Epochs | 2 | Epochs | 1 |
| Flash Attention | ✓ | Flash Attention | ✓ |

## E    MORE DETAILS ABOUT PROPOSED INPLAN3D BENCHMARK

In this section, we provide more details about InPlan3D. Each task contains a concise high-level instruction followed by a step-by-step breakdown of low-level actions, demonstrating the following characteristic:

- **Action-First Syntax**: Every step begins with a clear verb indicating the robots action.
- **Object and Attribute References**: Actions are directed toward specific objects, referenced both semantically (e.g., the central conference table) and structurally (*e.g.*, [table-0]).
- **Spatial and Contextual Cues**: Several steps include locational qualifiers(*e.g.*, beside the desk), helping localize the task in 3D space.

In Figure 9, we provide word cloud analyses of the Actions and Objects appearing in the dataset. As shown in Figure 10, most tasks in InPlan3D contain 4 to 6 steps with 30 to 60 words in total. Figure 11 presents several examples from InPlan3D.

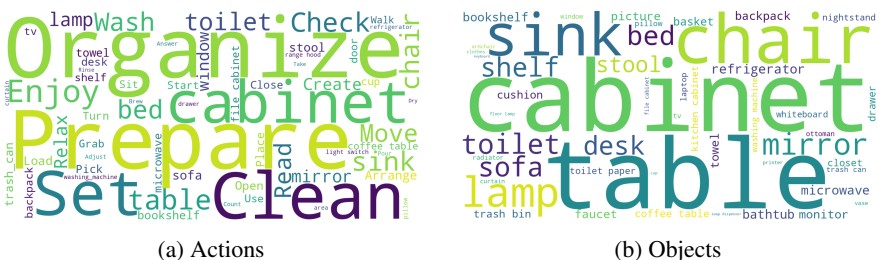

(a) Actions             (b) Objects

Figure 9: Wordclouds of (a) Actions and (b) Objects

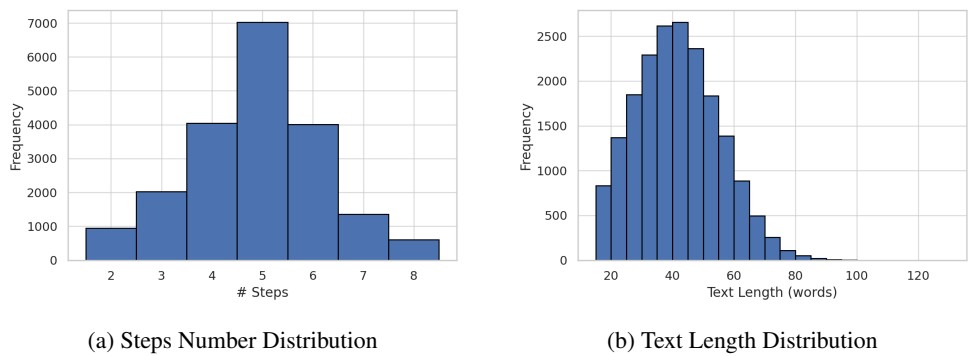

(a) Steps Number Distribution         (b) Text Length Distribution

Figure 10: Data Distribution of (a) Steps Number and (b) Text Length

**Task 1**

**Prepare the meeting room for a video conference.**

1. Walk to the central conference table. [table-0]

2. Adjust the chairs around the table. [chair-1]

3. Remove any clutter from the table surface. [table-0]

4. Clean the monitor on the wall. [monitor-19]

5. Close the door for safety. [door-17]

**Task 2**

**Restock supplies in the office.**

1. Walk to the supply shelf. [file cabinet-39]

2. Check inventory for stationery and files. [file cabinet-39]

3. Bringing new supplies for the storage area. [cabinet-36]

4. Arrange supplies neatly on the shelf. [file cabinet-39]

5. Close any open drawers or boxes. [cabinet-36]

**Task 3**

**Restock supplies in the office.**

1. Walk to the sink. [sink-45]

2. Rinse the dishes in the sink. [sink-45]

3. Clean the sink using soap and a sponge. [sink-45]

4. Dry the sink area with a towel. [sink-45]

**Task 4**

**Set up the study corner for working.**

1. Walk to the chair beside the desk. [chair-12]

2. Push the chair in and align it with the desk. [chair-12]

3. Arrange books and papers on the desk. [desk-13]

4. Turn on the nearby table lamp. [lamp-19]

Figure 11: Examples of InPlan3D benchmark.

