# OpenReview forum: "SITE: BRIDGING TEXT AND IMAGE MODALITIES WITH LLMS FOR 3D SCENE UNDERSTANDING"
_ICLR.cc/2026/Conference — ICLR 2026 Conference Withdrawn Submission_

### Official Review · Reviewer_o7i6 · 2025-10-25

**Soundness:** 2
**Presentation:** 3
**Contribution:** 2
**Rating:** 2
**Confidence:** 4

**Summary:**

This paper proposes SITE (Single-Image and Text Encoders), a framework aimed at integrating language and visual information for structured 3D scene understanding. The key idea is to use pretrained large language models (LLMs) and vision backbones to interpret image and text inputs jointly, and to produce object-level semantic scene graphs that can be queried or reasoned upon. The authors claim that SITE generalizes across multiple tasks including captioning, question answering, and scene parsing by leveraging textual annotations derived from GPT4Scene.

Experiments are conducted on some reasoning and grounding benchmarks, showing slightly improved or comparable performance to GPT4Scene. The authors argue that their method achieves better grounding between 2D and 3D semantics.

**Strengths:**

1. Use of structured representations: The idea of constructing a scene graph or symbolic structure for 3D understanding is meaningful for tasks like visual question answering and robotic navigation.

2. Readable presentation: The methodology and experiments are presented clearly, and the paper is reasonably well-organized with informative figures and visualizations (see Fig. 2 and 3).

3. Timely topic: Symbolic reasoning for 3D scene understanding is an important direction.

**Weaknesses:**

1. Limited novelty.
The proposed SITE framework mainly combines existing components from GPT4Scene and 3D-GRAND, such as object markers, BEV annotations, segmentation modules from Mask3D, and scene graph construction rules(such as 3D-GRAND). These external modules perform most of the core reasoning, leaving SITE’s contribution marginal.

2. Hidden dependencies.
The paper downplays its reliance on GPT4Scene’s annotations and Mask3D segmentation, even though these are essential to SITE’s performance. The claim of “no point encoder” in Table 2 is misleading because Mask3D is still used for object-level information.

3. Marginal performance gain.
SITE achieves only very small improvements (0.3–0.5 points) over GPT4Scene, which are within the normal variance range. This shows limited practical value and weak empirical justification.

4. Unclear 3D connection.
Despite claiming to enable “3D scene understanding,” SITE only processes image and text inputs without directly modeling 3D geometry. The title and claims therefore overstate the actual capability.

5. Missing implementation details.
The paper also omits key training and architecture details (e.g., fusion process, loss design, encoder setup), which hurts reproducibility and makes it unclear what part of the system truly drives performance.

**Questions:**

1. Could the authors clarify which parts of GPT4Scene’s pipeline are reused directly (object markers, BEV maps, segmentation annotations)? How much training data or annotation effort depends on GPT4Scene?

2. What is the input prompt of LLMs in training? From my view, previous methods, such as Video 3D-LLM and GPT4Scene, directly take the question and 3D tokens as the input, while the proposed method takes the structured language from scene parsing as the input. The author does not discuss this difference in the paper.

2. Table 2 claims “no point encoder” is used, yet Mask3D is mentioned in Appendix B as part of the preprocessing. How do you reconcile this inconsistency?

3. What exactly differentiates SITE’s Single-Image and Text Encoder from standard CLIP-like dual encoders?

4. How is “bridging text and image modalities” equivalent to “3D scene understanding”? Is there any component of SITE that directly models 3D geometry?

5. Can you provide quantitative or qualitative examples where SITE improves interpretability or scene parsing compared to GPT4Scene?

---

### Official Review · Reviewer_htT2 · 2025-10-30

**Soundness:** 3
**Presentation:** 3
**Contribution:** 3
**Rating:** 6
**Confidence:** 3

**Summary:**

This paper introduces SITE, an end-to-end framework that uses only BEV image and textual description as input for 3D scene understanding. The proposed method is able to achieve superior performance across various task including 3D-QA, grounding and captioning. The paper also introduces a more challenging benchmark that considers long-horizon task planning in 3D scene, where the proposed model also lead the performance over previous works.

**Strengths:**

1. The paper introduces a 3D scene understanding model without 3D input, which is an interesting direction to approach.

2. The paper demonstrates that even without 3D input, the model is still able to perform well across various tasks. Especially on the most challenging setting InPlan3D, the model is able to outperform other models by a large margin.

3. The InPlan3D benchmark would be useful for further study that bridges embodied AI and 3D vision.

**Weaknesses:**

1. Based on my understanding, the textual input to SITE requires some preprocessing from the 3D scene, which might be time-consuming if using it online.

2. The dataset introduced in LEO [1] also covers some task planning aspect in 3D scene, can the author clarify the difference between the proposed InPlan3D and their dataset?

3. Lack of analysis why text-and-image only input model can be better when it comes to long-horizon task planning.


[1] Jiangyong Huang, Silong Yong, Xiaojian Ma, Xiongkun Linghu, Puhao Li, Yan Wang, Qing Li, Song-Chun Zhu, Baoxiong Jia, and Siyuan Huang. An embodied generalist agent in 3d world.arXiv preprint arXiv:2311.12871, 2023d.

**Questions:**

1. Can the author provide some intuition why the sentence selection block needs image as an additional selection information source? Why is visual signal useful for sentence selection if the sentences are derived from the scene?

---

### Official Review · Reviewer_jJMM · 2025-11-02

**Soundness:** 3
**Presentation:** 2
**Contribution:** 2
**Rating:** 6
**Confidence:** 4

**Summary:**

The paper proposes SITE, a framework that bridges 3D scene understanding and large language models by introducing a Scene2Text module and a Sentence Selection block. The key idea is to convert 3D environments into structured textual scene information (object attributes and spatial relations) combined with a single BEV image, which are then fed into an LLM for downstream tasks such as 3D question answering, grounding, and task planning. Experiments on multiple benchmarks (ScanRefer, ScanQA, SQA3D, and the new InPlan3D) demonstrate competitive or superior performance compared to previous 3D-LLMs.

**Strengths:**

1. The paper introduces a novel form of scene representation -- converting structured 3D scene graphs into natural language “scene information.” This design effectively aligns 3D perception with LLM-friendly textual inputs.

2. The Sentence Selection module is practical and well-motivated. It improves efficiency and relevance by filtering the most question-related sentences.

3. The method achieves strong performance across diverse benchmarks with significantly reduced visual input cost.

**Weaknesses:**

- The Scene2Text pipeline involves multiple heavy steps (3D reconstruction, instance segmentation, LLM-based captioning, and self-reflection). While this is acceptable for offline data preparation, in online inference scenarios the system would still need to generate the textual scene description on-the-fly. How long does Scene2Text take to process one scene or one ScanNet sample? Some quantitative timing or computational cost analysis is needed to assess its practical applicability.

- The current experiments mainly validate downstream task performance, but it remains unclear how accurate the generated Scene Information itself is. Since datasets like ScanNet have ground-truth object masks and annotations, the paper could report metrics such as IoU, precision/recall of relations, or show visual comparisons between generated and GT scene graphs / captions to support the reliability of Scene2Text outputs.

**Questions:**

See Weaknesses

---

### Author Response · Authors · 2025-11-17

We sincerely appreciate the time and effort that the reviewers and the AC have devoted to evaluating our submission. After carefully considering the current reviews and scores, we have decided to withdraw the manuscript from further consideration.

---

### Note · Authors · 2025-11-17

I have read and agree with the venue's withdrawal policy on behalf of myself and my co-authors.